# Advances in Understanding Microbial Deterioration of Buried and Waterlogged Archaeological Woods: A Review

Adya P. Singh [1], Yoon Soo Kim [2,*] and Ramesh R. Chavan [3]

1 Scion (New Zealand Forest Research Institute), Rotorua 3046, New Zealand; adyasingh@hotmail.com
2 Department of Wood Science and Engineering, Chonnam National University, Gwangju 61186, Korea
3 School of Biological Sciences, University of Auckland, Auckland 1442, New Zealand; r.chavan@auckland.ac.nz
* Correspondence: kimys@jnu.ac.kr

**Abstract:** This review provides information on the advances made leading to an understanding of the micromorphological patterns produced during microbial degradation of lignified cell walls of buried and waterlogged archaeological woods. This knowledge not only serves as an important diagnostic signature for identifying the type(s) of microbial attacks present in such woods but also aids in the development of targeted methods for more effective preservation/restoration of wooden objects of historical and cultural importance. In this review, an outline of the chemical and ultrastructural characteristics of wood cell walls is first presented, which serves as a base for understanding the relationship of these characteristics to microbial degradation of lignocellulosic cell walls. The micromorphological patterns of the three different types of microbial attacks—soft rot, bacterial tunnelling and bacterial erosion—reported to be present in waterlogged woods are described. Then, the relevance of understanding microbial decay patterns to the preservation of waterlogged archaeological wooden artifacts is discussed, with a final section proposing research areas for future exploration.

**Keywords:** waterlogged archaeological woods; wood deterioration; cell wall degradation; soft rot; bacterial tunnelling; bacterial erosion

## 1. Introduction

In nature, wood can deteriorate from microbial attack. While this is important for the recycling of carbon stored in the wood cell wall, it is a cause of enormous economic losses due to the deterioration of wooden structures built for human use. In outdoor environments, basidiomycete fungi, which cause white and brown rot of wood, play a dominant role in wood decomposition [1]. However, wood can also be attacked by soft-rot fungi and bacteria often under conditions that discourage the growth and activity of white and brown rot fungi. Wood exposed to high moisture conditions is not generally attacked by basidiomycete fungi [2] owing to the saturation of wood tissues with water. However, under these conditions wood is not prevented from attack by soft-rot fungi and bacteria, for example, timbers placed in cooling towers and in retaining walls [3–5]. These microorganisms are slow degraders compared to white and brown rot fungi, and therefore the timbers placed in moist and wet environments have a longer service life.

Waterlogging of wood occurs when wood is exposed to water-saturated or aquatic environments. Partial waterlogging can support the activity of soft-rot fungi and wood degrading bacteria [6]. However, when exposure conditions become anoxic due to complete waterlogging, for example, in deep ocean waters, ocean, river and lake sediments, and mud, wood is mainly degraded by erosion bacteria [7–18]. Soft-rot fungi and tunnelling bacteria may also be present but are much less frequent [13,14,16,18–20]. Erosion bacteria are regarded as the wood degrading microorganisms most tolerant to depleted oxygen

concentration. Wood degradation by erosion bacteria under anoxic conditions is also extremely slow. It is therefore not surprising that buried and waterlogged wooden structures have been found to survive hundreds and even thousands of years of exposure to such adverse environments. Although the abiotic deterioration of wood exposed to anoxic burial environments over long periods can also take place, abiotic factors are considered to play a minor role [21]. Well preserved ancient waterlogged wooden structures, such as sunken ships and their contents, are precious resources because they can inform us about the past civilisation as well as the environmental conditions of the time [22]. It is therefore important to preserve or restore such valuable artefacts in a condition that can provide us with such information. This requires detailed knowledge of the cause of their deterioration that may have occurred over prolonged exposure to anoxic conditions resulting from waterlogging, and the physical and chemical state of the excavated wooden objects [23]. This review will begin with a brief account of the chemical and ultrastructural characteristics of wood cell walls and a background of the micromorphological features associated with microbial degradation of lignocellulosic cell walls causing deterioration of buried and waterlogged archaeological woods before discussing the relevance of understanding decay patterns to the preservation of waterlogged archaeological wooden artefacts.

## 2. Chemical and Ultrastructural Characteristics of Wood Cell Walls

The type of wood and macro, micro and ultrastructural organisation of wood tissues in combination with the chemical characteristics of cell walls can be related to the manner and speed at which microorganisms degrade wood, particularly soft-rot fungi and bacteria which can attack water-saturated and waterlogged woods. Therefore, a brief account of the chemical and ultrastructural characteristics of wood cell walls, which are the primary source of nutrients for these microorganisms, is necessary in order to meaningfully understand the degradation processes associated with the microbial activity.

Forest trees can be distinguished into two main types: softwood and hardwood trees. They differ in the cellular composition of wood and also in the proportion of cell wall chemical constituents [24]. Whereas a large proportion of softwood consists of one cell type (tracheids), different cell types (vessels, fibres, fibre-tracheids) make up the hardwood. While at the ultrastructural level the cell wall of these cell types is remarkably similar, subtle differences occur in the chemical composition.

### 2.1. Cell Wall Composition

The cell wall is composed of three main polymers—cellulose, hemicellulose and lignin [24], with cellulose being the most dominant component. Generally, softwoods have a higher cellulose content (40–50%), greater amount of lignin (26–34%) and lower amount of hemicelluloses (7–14%) compared to hardwood species (cellulose 38–49%, lignin 23–30%, hemicelluloses 19–26%). Extractives are an important component of heartwood [25,26] and are most abundant in tropical woods, and serve to enhance wood durability [27]. Softwoods and hardwoods also differ in the type of lignin present in their cell walls [24]. Softwood tracheid walls mainly consist of guaiacyl lignin and those of hardwood cells consist of syringyl lignin, except vessel cell walls which contain both guaiacyl and syringyl lignin. Cellulosic chains form stiff cables (microfibrils) which can be readily visualised under an electron microscope. There is a much smaller amount of pectin also present, which is primarily located in the middle lamella [28]. Wood-degrading microorganisms (soft-rot fungi and bacteria) that are present in water-saturated and waterlogged woods can readily depolymerise cellulosic and hemicellulosic components but differ in their ability to modify lignin, a recalcitrant component of the cell wall. Therefore, it is important to have knowledge of the concentration and distribution of lignin in various cell wall regions and tissues in order to understand the degradability of different cell wall layers/structures (e.g., vestures, warts), wood types (hardwood, softwood, normal wood, reaction wood) and cell types [18,29].

### 2.2. Cell Wall Formation and Ultrastructure

A brief description of the formation of the cell wall is considered not to be out of place here, as it can shed light on the relationship between primary and secondary cell walls and the sequence that follows in their development. During the expansion of cambial derivative cells, destined to become secondary xylem, the primary cell wall is deposited over the middle lamella, which consists largely of pectic polymers [28] and develops from the cell plate formed during the cytoplasmic division (cytokinesis) between divided cells. The cellulosic component of the cell wall deposited subsequent to the establishment of the middle lamella is generated from the activity of cellulose synthase complexes embedded in the plasma membrane. These complexes produce linear chains of cellulose which are hydrogen bonded giving rise to a fibrillar structure, referred to as the microfibril, which can be visualised under an electron microscope as 2–5 nm rods. The timing and sequence of deposition of the hemicellulosic component are poorly understood. It is assumed that the deposition of cellulose and hemicellulose is closely coordinated [30], the hemicellulosic component forming structural links (bridges) between microfibrils, which have been visualised by a range of high-resolution tools and techniques [31–34]. As the primary cell wall material is deposited during cell expansion growth, the cell wall is continually modified to allow coordinated deposition of cell wall materials and maintenance of cell wall integrity. The secondary wall is deposited at the conclusion of expansion growth of differentiating secondary xylem cells, forming a three-layered structure.

Since cells are no longer expanding at this stage, microfibrils in the secondary cell wall maintain an orderly disposition, remaining parallel to one another. The plywood-type organisation of the secondary cell wall in which microfibrils differ in their orientation in successive layers (S1, S2, S3) (Figure 1) is designed in a way that confers the wood cell wall its optimum strength and stiffness. Recent secondary cell wall models propose the inclusion of a transition layer/zone between the secondary layers because electron microscopic images obtained from various regions across the cell wall demonstrate that the angle of microfibrils in S1–S2 and S2–S3 interfacial regions gradually changes [35]. This type of cell wall design undoubtedly helps prevent fracturing in the interfacial region between cell wall layers under stresses generated from internal and external factors.

Lignin, the other major component of wood cell walls, is incorporated subsequent to the establishment of the cellulosic-hemicellulosic infrastructure, initially beginning in the middle lamella and progressing towards the inner cell wall. However, reports of atypical lignification suggest that lignification may not always initiate in the middle lamella [36]. Although biochemical aspects of the synthesis of lignin monomers and their transformation into lignin polymer have been extensively investigated, opinions differ as to whether or not the initial deposition of lignin monomers in the cell wall is regulated. Lignin monomers are considered to passively enter into the spaces within the cellulose–hemicellulose complex, and indications are that the depositing lignin is constrained by parallel oriented cellulose microfibrils [37], as electron microscopic images of lignin lamellae parallel to microfibrils have been captured. Lignin in the cell wall occurs in a complex relationship with the carbohydrate component, and opinions vary for the exact nature of chemical bonding. However, it is important to understand the nature of the interaction from cell wall biodegradation perspectives. The microorganisms that degrade buried and waterlogged archaeological woods apparently possess a capacity to unlock the recalcitrant lignin from the polysaccharide to gain access to the latter. Understanding how they accomplish this requires a knowledge of the nature of carbohydrate–lignin linkages and the enzymes/radicals that microorganisms deploy. This is an area open for future advances to be made. Nishimura et al. [38] have provided evidence for covalent bonding between lignin and carbohydrates; however, knowing from recent advances that lignin in the cell wall interacts with hemicelluloses via electrostatic interactions and occurs in its own nano domain [39], and can rearrange itself when wood is subjected to compression load [40], further developments are needed to fully understand the nature of lignin–carbohydrate interactions.

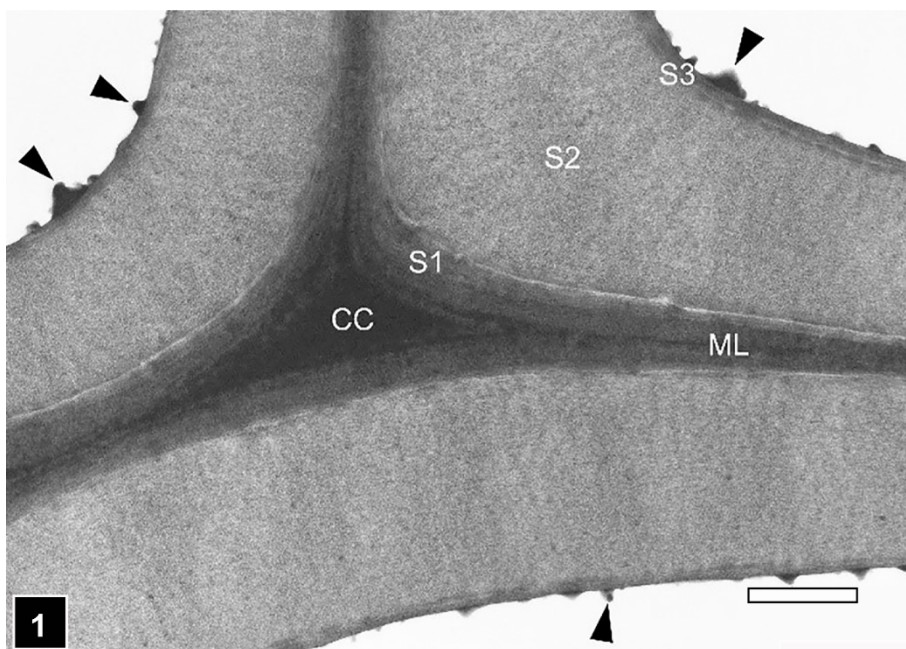

**Figure 1.** TEM of a transverse section through silver fir (*Abies alba* Mill.) tracheids. The secondary cell walls display a lamellar organisation. CC, cell corner; ML, middle lamella; S1, S2, S3, secondary wall layers; warts (arrowheads). Scale bar = 1µm. The image is courtesy of Prof. Jong Sik KIM, Chonnam National University, South Korea.

The secondary cell wall layers not only differ in microfibril orientation, with microfibrils oriented axially in the S2 layer and perpendicular to this in the S1 and S3 layers, but also in the thickness, and sometimes in the concentration of lignin. The S2 layer, being the thickest, most noticeably in hardwood fibres and latewood tracheids of softwoods, is the most dominant part of the wood cell wall and thus a rich source of nutrients for the microorganisms present in waterlogged woods. Variability in lignin concentration among the secondary cell wall layers is worthy of special attention because of the high resistance of lignin to microorganisms degrading wood in wet environments [15].

Cell walls of certain tissues consist of more than the usual three layers. Fibre cell walls in certain hardwoods, such as *Homalium foetidum* ((Roxb.) Benth.) [41] and kempas (*Koompasia malaccensis* (Maingay)) [42], are composed of multiple layers (multilamellar cell walls). This unique cell wall design optimises cell wall mechanical properties [43], particularly of fibre walls consisting of alternating thick and thin lamellae with differing orientations of microfibrils. Usually, the thick and thin lamellae also differ in lignin concentration, with thin lamellae displaying a greater concentration of lignin compared to thick lamellae, as revealed by imaging of KMnO4 stained ultrathin sections by TEM [41,42]. The cell walls of fibres in certain plants which do not form wood tissues, such as bamboo [43], are also multilamellar [44]. Such fibres likely provide protection to thin-walled tissues, such as thin-walled parenchyma, which are susceptible to collapse under physical and mechanical loads imposed by wind and other external factors.

Knowledge of this type of cell wall feature is important also from the perspective of microbial degradation of cell walls. An excellent example of this is found in TEM images of multilamellar fibre cell walls attacked by soft-rot fungi, where thick lamellar regions of the cell wall are extensively degraded but lignin-rich thin lamellae are relatively resistant, as the half-moon shape of soft-rot cavities present bordering thin lamellae would suggest [41,42]. Soft-rot fungi are present in both terrestrial and aquatic environments, and therefore this knowledge is relevant also to wood degradation in waterlogged environments.

The relationship between cell wall ultrastructure and lignin-rich wood structures to microbial degradation has recently been described [18,29], the knowledge of which is relevant to waterlogged woods. The investigated features are: microfibril orientation, cell wall regions with high lignin concentration, particularly the middle lamella, initial pit borders [18,45], vestures and warts [18,46–48], tyloses [49,50], highly lignified ray tracheids [18,47,51] outer S2 wall of compression wood [52–55], phenolic deposits in parenchyma cells and other wood tissues [56–58].

Knowledge of the micromorphological patterns produced by wood degrading microorganisms is important for recognising which types of microorganisms cause degradation of buried and waterlogged archaeological woods. Based on the images obtained using light and electron microscopy, three different types of microorganisms have been implicated in the deterioration of such woods: soft-rot fungi (SR) producing cavities in the cell wall (type I soft rot), tunnelling bacteria (TB) and erosion bacteria (EB). It is important in this context to emphasize that while the majority of studies of waterlogged woods have reported EB as the main degraders of lignocellulosic cell walls, others have also found the presence of SR and TB; albeit the latter two types less frequently [reviewed in 14,15]. Therefore, a brief description presented of the micromorphology of the degradation patterns produced by the three types of microorganisms and the advances made that led to the knowledge gained will serve as an important diagnostic base for those investigating wood degradation in waterlogged environments.

## 3. Fungal Degradation

Soft-rot fungi cause two well-defined patterns of wood degradation, which have been described as type I and type II. Type I is characterised by cavity formation within the cell wall during the degradation process. In type II, the cell wall is eroded by fungal hyphae present in the cell lumen. In our review, the micromorphological pattern of only soft rot type I is presented, as it has been reported that type I soft rot is exclusively present in waterlogged archaeological woods.

*Cavity-Forming Soft Rot (Type I Soft Rot)*

Fungi causing type I soft rot are present in a wide range of terrestrial and aquatic environments [2,19,59]. Some species have even adapted to degrade wood under extreme conditions, such as those present in Arctic [60] and Antarctic [61,62] regions. Cavities in cell walls are produced by SR belonging to Ascomycetes, although some white-rot fungi have been reported to also produce cavities [59]. SR are more common in moist/wet environments which discourage the growth and activity of the aggressive white and brown rot fungi, and where they often coexist with wood-degrading bacteria [3,5,6,45,62,63].

The decay pattern (cavities in the cell wall) produced by SR can be readily recognised using a light microscope (LM) [64] (Figure 2), which provides sufficient high resolution to follow fungal pathway within wood tissues and obtain details on the micromorphology of forming and developed cavities, as well as to assess the orientation of cavities relative to cell wall microfibrils. Because LM also enables rapid evaluation, it has been widely used as a diagnostic tool and to study processes associated with wood degradation by SR. Further advances using SEM and TEM, which offer much greater resolution compared to LM, have yielded valuable additional information on the processes of soft-rot cavity formation and the ultrastructure of cavity-forming fungal hyphae [65], presence of a granular material within cavities (Figure 3), considered to represent a mixture of slime, melanin and modified lignin residues [59] and in some cases presence of a wider irregular zone around cavity-forming hyphae [66] (Figure 3) compared to the usual concentric form of cavities observable in transverse sections [29,67] (Figures 2 and 4).

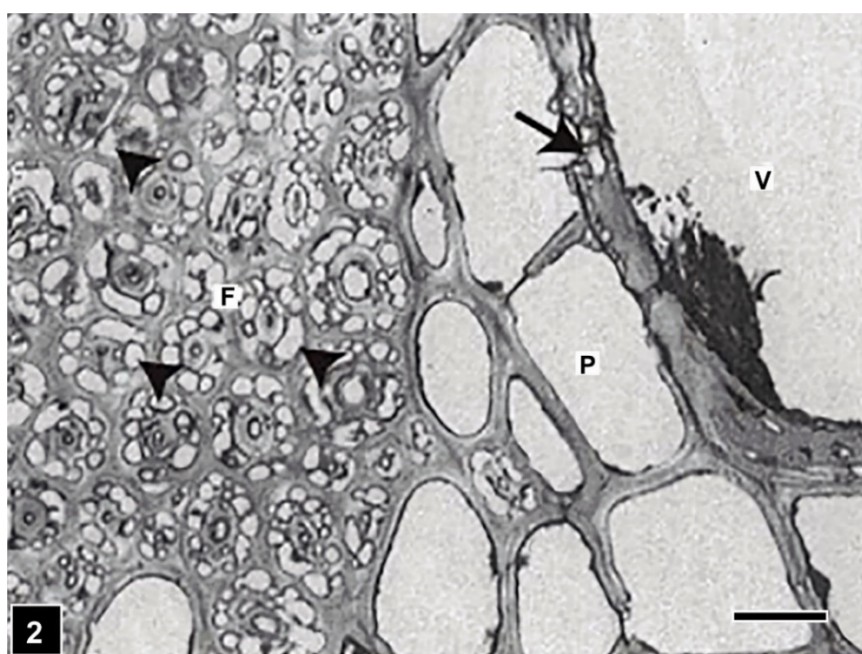

**Figure 2.** LM of a transverse section through *Koompassia malaccensis* wood attacked by soft rot. Many soft-rot cavities are present in fibre (F) walls (arrowheads), but only a few in the vessel (V) wall (arrow). P, parenchyma. Scale bar = 20 µm. The image is reproduced from Singh et al. (2018) IAWA J.

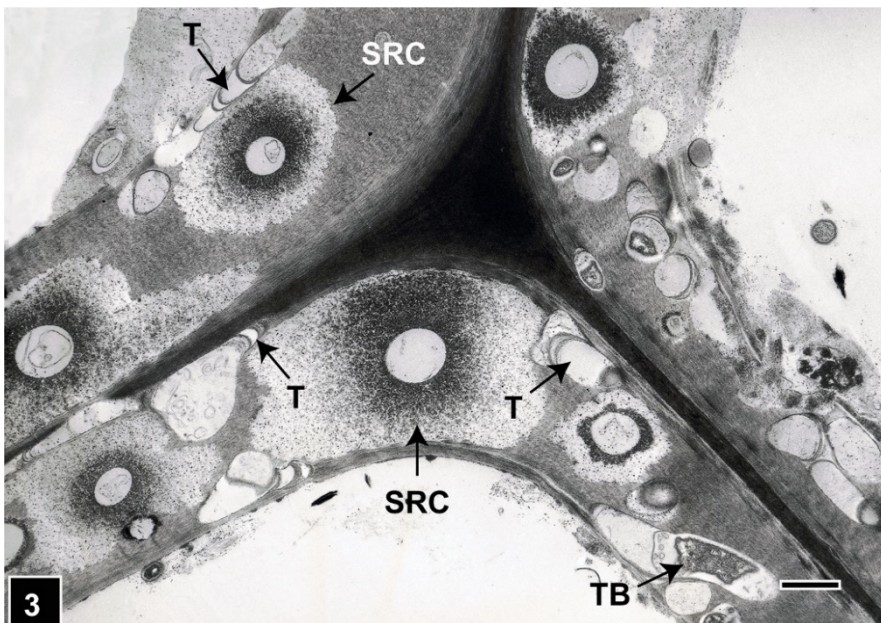

**Figure 3.** TEM of a transverse section through *Pinus radiata* tracheids attacked by soft-rot fungi and tunnelling bacteria. Soft-rot cavities (SRC) display a diffuse degradation pattern. Tunnelling bacteria (TB) and tunnels (T) are present in cell wall regions not occupied by soft-rot cavities. All cell wall regions, including the highly lignified middle lamella and S3 layer, are degraded by TB. Scale bar = 2 µm. The image is reproduced from Singh et al. (2019) IAWA J.

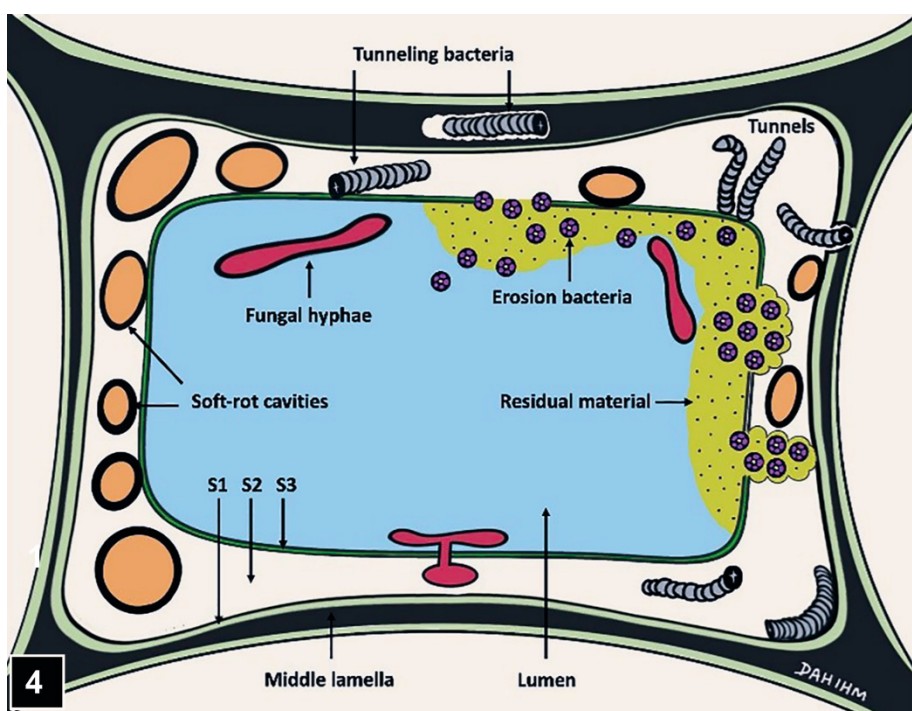

**Figure 4.** A diagram showing the micromorphological patterns produced during attack of a tracheid wall by soft-rot fungi and tunnelling and erosion bacteria. The diagram from Kim and Singh (2000) IAWA J. is kindly re-drawn by DahIhm Kim.

The process of cavity formation has been reviewed by several workers, including Daniel and Nilsson [59] and Daniel [68]. Briefly, the process involves penetration of the cell wall by hyphae colonising the cell lumen. The penetrating hyphae align themselves with cellulose microfibrils, following L-bend or T-branching, a feature considered to be a pre-requisite for cavity initiation, as hyphal alignment triggers enzyme production. In longitudinal sections of wood tissues, cavities are seen to run parallel with microfibrils [69–72]. Cavities develop from the degradation of the wood cell wall around the hyphae. Cavities appear diamond shaped in longitudinal sections of the cell wall, and circular or near-circular when the cell wall is sectioned transversely. These features enable detection and confirmation possible of the presence of SR type I attack in decaying wood, including waterlogged wood. The composition of the cell wall influences cell wall degradation, particularly the type and concentration of lignin [59,68,73]. This is apparent from microscopic observations showing resistance of lignin-rich middle lamella [59,67,74] and the S3 layer (Figure 3) [67], particularly in softwoods where the S3 layer is often more highly lignified than S1 and S2 layers [75]. Microscopic studies have also provided evidence of the resistance of the highly lignified ray tracheids [51] and initial pit borders in conifers [45]. Support for lignin influence on cavity formation also comes from TEM images showing the presence of half-moon-shaped cavities in multilamellar cell walls, containing thick and more highly lignified thin lamellae. The face of the cavities in contact with the thin lamellae has a flattened appearance (Figure 5), suggesting that the development of the usual circular form of cavities is constrained [41,42]. The type of lignin also has an influence on cavity formation, with guaiacyl lignin being more resistant than syringyl lignin [73]. Delay in cavity formation in guaiacyl lignin-rich vessel cell walls [59] is supportive of this view. Indications are that there may also be an effect of physical constraint on cavity formation. For example, cavities generally form in the S2 layer but are rare in the extremely thin S1 layer. However, cavities can develop in the S1 layer of compression wood, where S1 is wider than in the tracheids of normal wood, which suggests that there may be a requirement for a minimum width of the cell wall layer for cavity formation.

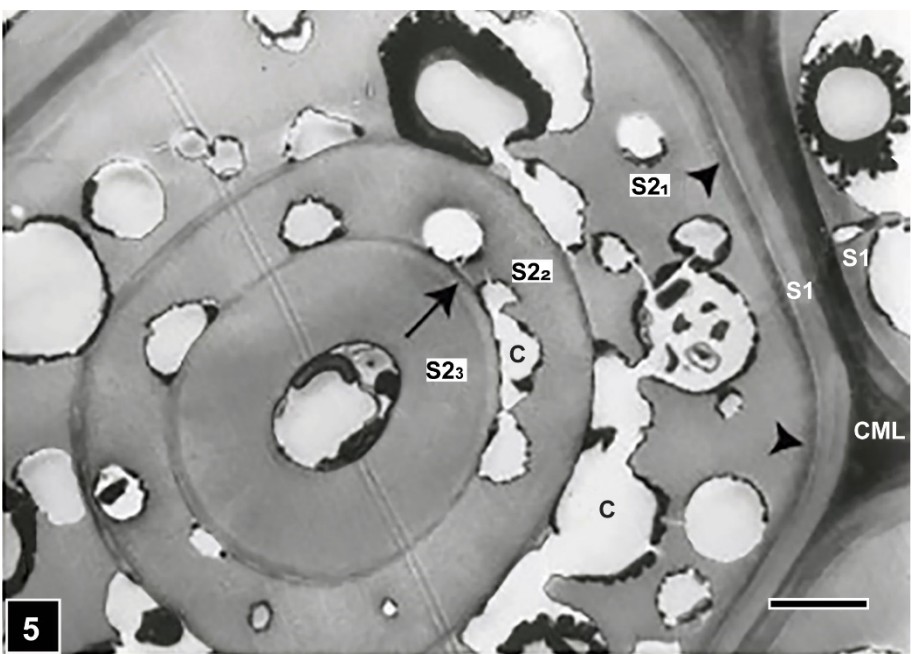

**Figure 5.** TEM of a transverse section through a *Koompassia malaccensis* fibre wall attacked by soft rot. The S2 wall is multilamellar, consisting of thick lamellae (S2-1, S2-2, S2-3) alternating with extremely thin lamellae (arrow). The face of soft-rot cavities (C) along the more highly lignified thin lamellae has a flattened appearance. The arrowheads point to a highly lignified region of the S2-1 lamella, underlying S1. CML, compound middle lamella. Scale bar = 2 μm. The image is reproduced from Singh et al. (2018) IAWA J.

Less is known about whether cavity-forming hyphae produce diffusible enzymes; some investigations support the diffusible nature of the enzymes produced [76]. In the majority of cases, cavities formed have a well-defined border, which suggests that the enzymes produced by wood-degrading fungi cannot diffuse into the surrounding sound wood cell wall, as it is believed that fungal enzymes are too large to penetrate the nanostructure of intact cell walls [77]. For brown rot fungi, it has been proposed that in addition to producing enzymes, these fungi deploy a non-enzymatic system [77,78] consisting of small molecular substances which can modify the cell wall, enabling cellulolytic and hemicellulolytic enzymes to gain entry into the cell wall and access holocellulosic components. It is not known whether SR also produce a non-enzymatic diffusible system, but it is a distinct possibility at least for those fungi causing diffuse degradation [66,67] (Figure 3), where cell wall dissolution extends well beyond the cell wall regions where hyphae are present. The compositional changes due to cell wall degradation by SR have been reported by several workers [reviewed in 59]. Proportionately much greater losses incur in holocellulosic components compared to lignin. Some workers have reported significant losses in lignin for some ascomycete fungi [73,79,80]. However, in all cases, holocellulose is preferentially degraded.

## 4. Bacterial Degradation

### 4.1. Developments Leading to Confirmation That Certain Bacteria Can Degrade Lignified Wood Cell Walls

Bacterial presence in decaying wood has long been recognised [81]. Early studies aimed to understand whether bacteria can degrade sound wood employed LM to examine decaying wood from natural environments and wooden constructions in service [82]. While bacterial presence in decaying wood was confirmed and decay features that did not resemble those described for wood-degrading fungi were observable, the progress in understanding whether bacteria could degrade lignified cell walls was hampered by the inability to obtain detailed views of such patterns due to the limited resolution of

LM. Co-existence of bacteria with fungi in wood-decaying natural environments was also a complicating factor. Furthermore, attempts to produce wood decay by pure or single bacterial isolates were not successful [7]. This led to the belief that bacteria alone could not degrade lignified cell walls and only played a minor role in wood decay. Later studies also reported the inability of bacterial strains to degrade lignocellulosic cell walls under laboratory conditions [8]. The application of SEM provided greater insights into the micromorphology of the unusual decay [7,83–85]. However, it became possible to unequivocally confirm that bacteria can degrade lignified cell walls only when TEM was employed, which made it possible to examine extremely thin (ultrathin) sections of polymer-embedded decaying wood tissues at high resolution, providing detailed features of cell wall degradation and the spatial relationship of bacteria with cell wall regions being degraded [29,47,74,86]. Furthermore, the application of potassium permanganate (KMnO4), a reagent used as a fixative or stain to contrast lignin in plant and wood cell walls [87–90] prior to examination of ultrathin sections with TEM provided useful information on the degradation of wood cell wall regions and structures varying in lignin concentration [29,86]. TEM examination of KMnO4 stained ultrathin sections provided high definition images, revealing features of bacterial morphology and ultrastructure, bacterial association with decaying cell wall regions and the fine structure of the distinctive decay patterns produced. The detailed information obtained enabled the bacterial degradation patterns to be placed into two well-defined categories, which were named tunnelling and erosion and the bacteria producing those tunnelling bacteria (TB) and erosion bacteria (EB) [59,74,86].

### 4.2. Tunnelling Type Bacterial Degradation

4.2.1. Environments and Wood Substrates

Following confirmation, using TEM, that certain bacteria can degrade lignified cell walls by way of tunnelling within the cell wall, it became possible to recognise this type of bacterial decay also using LM and SEM, which led to a flurry of activities leading to the reports of wide presence of tunnelling degradation of wood in terrestrial as well as aquatic environments [59,74,86], including waterlogged archaeological woods [15,16,20,52,62] from different parts of the world. However, tunnelling type attack is most common in wood in contact with moist soils and exposed to wet environments, conditions unfavourable to the aggressive white and brown rot fungi. However, such conditions also support the activity of SR and EB, and thus mixed attacks on wood by SR, TB and EB have been reported [3,6,45,54,63,67,91], including waterlogged archaeological woods [16,20,62]. Moreover, when present in waterlogged archaeological woods, tunnelling type attack is generally confined to outer tissue layers of wooden objects. This and the observation that shipwrecks recovered from deep ocean sediments, where conditions can be anoxic, were found to be almost exclusively attacked by EB, suggest that the attack of TB on sunken ships is likely to have occurred prior to or during submergence of ships when the ocean water was sufficiently oxygenated to support the activity of TB, as these bacteria are considered to require oxygen for the degradation of lignocellulosic cell walls. The presence of TB attack in shipwrecks from intertidal sites [19] and wood from coastal waters of Antarctica [62] has been reported.

TB can tolerate conditions considered extreme for other wood-degrading microorganisms, particularly the highly destructive white and brown rot fungi. For example, TB attack is often associated with wood products that have been placed in service in ground contact after treatment with the copper-chrome-arsenate (CCA) preservative at retentions high enough to discourage attack by white and brown rot fungi [59,74,86]. TB can also attack high lignin wood species and heartwoods containing extremely high levels of toxic extractives [56].

4.2.2. The Micromorphological Pattern Associated with Tunnelling Type Attack: Relevance to Waterlogged Archaeological Woods

Understanding the micromorphological pattern produced by TB during degradation of lignocellulosic cell walls is important in order to be able to recognise the presence/absence of bacterial tunnelling of cell walls in waterlogged archaeological woods. The pattern is unique and distinctive and cannot be mistaken for any other type of microbial degradation. When examined, particularly under TEM, TB type degradation can be readily identified even in advanced stages of cell wall degradation, based on the ultrastructural morphology of tunnels, which is very relevant to waterlogged archaeological wooden artefacts, where wood tissues may be severely degraded. Remarkably, the basic tunnel micromorphology is consistent regardless of wood type and exposure conditions, with only minor variations [29,62].

TB are rod-type Gram-negative bacteria, judging by the presence of a membranous enclosing cell wall. However, TB are capable of changing their shape (pleomorphic) [59,86], particularly while encountering physical and chemical constraints during their movement within the cell wall. For example, TB often assume a dumbbell shape as they traverse the highly lignified middle lamella with the constricted part of the bacterial cell observable within the middle lamella, and become slender and elongated when present in the S1 layer, a very thin part of the secondary cell wall [91]. TB are non-flagellate and move as they glide on a slimy material (likely a mucopolysaccharide) they extrude from their surface. The morphology and ultrastructure of TB and the stages of bacterial entry into the cell wall from the cell lumen have been reviewed [74,86,92]. The most important feature that serves as a diagnostic signature for TB type degradation is the presence of tunnels within the cell wall [91,93], even when tunnels are not intact and only their remnants are present, such as in heavily degraded waterlogged archaeological woods [20].

Briefly, after colonising the lumen of wood cells, TB attach themselves to the luminal face of the cell wall (exposed face of the S3 layer) with the help of the extracellular slime they produce during attachment and throughout the process of tunnelling within the cell wall. When in contact with the S2 layer, TB preferentially tunnel through this region of the cell wall, which is the thickest part of the cell wall and contains the bulk of cell wall constituents, although TB have the capacity to degrade all cell wall regions, including the highly lignified middle lamella [59,67,91] and the S3 layer [29,67] (Figures 3 and 4), which is an extremely thin layer of the cell wall and in some conifers is also highly lignified, such as in *Pinus radiata* [75]. TB degrade the cell wall as they glide on the extracellular slime, and in the process, tunnels closely fitting the circumferential dimension of these bacteria are produced. The micromorphological pattern of degradation suggests that TB are able to move in all directions within the cell wall (Figures 4, 6 and 7), and unlike cavity-forming soft-rot fungi which align themselves with microfibrils, TB movement is not constrained by the orientation of microfibrils [59,74,86]. Although observations showing a correspondence between microfibril orientation and the direction of tunnelling within the S1 layer [91], an extremely thin layer of the secondary cell wall (Figure 8), suggest that there may be some level of physical constraint or some degree of bacterial preference for microfibril orientation. In addition to middle lamella and the S3 layer, TB have been reported to degrade other highly lignified cell wall structures, such as the initial pit border [18,45,52] and the outer S2 wall in compression wood [53,55], which suggests that lignin, a recalcitrant cell wall polymer, is not a deterrent for TB. However, the infrequent presence of tunnels in the cell corner middle lamella, the most highly lignified region of the conifer cell wall, argues in favour of TB's preference for cell wall polysaccharides over lignin. The single most important feature of tunnelling type degradation is the morphology and ultrastructure of tunnels, which reveals the direction of TB movement within the cell wall and serves as a diagnostic signature for the presence of tunnelling type degradation even in the most advanced stages of cell wall degradation and in situations involving mixed microbial attacks in water-saturated wooden structures [29,45,67,91] and waterlogged archaeological woods [12,20,62]. As TB degrade the cell wall and advance, they leave a trail of slime

behind as a part of the tunnel. Thus, there is always a sheath of slime around TB and in contact with the surrounding cell wall. The most intriguing features of the tunnel are captured by TEM, which reveal the presence of crescent-shaped periodic bands (likely discs in 3D) compartmentalising the tunnel [91–94] (Figure 9). It has been assumed that the slime is continually extruded from the bacterial surface, and the bands reflect tunnel regions where the slime becomes most highly concentrated around the posterior of TB (Figure 9), as these bacteria stop to perhaps replenish their enzymes [95], assuming a shape (crescent shape) corresponding to the dome shape of the posterior of the bacterial cell [91]. The band concavity always faces the direction in which TB move (Figures 6 and 9), and thus even when bacteria are missing in sectional views or are absent from the tunnel, particularly in extensively degraded regions of the cell wall, where only scant tunnel bands may be present, the direction of TB movement can be readily assessed. This feature is also applicable to waterlogged archaeological woods in determining the presence/absence of TB attack. In situations of mixed microbial attacks present within the same wood cell wall, the tunnel bands and their remains also inform us of the nature of the spatial relationship between the co-existing microorganisms, such as that described for soft rot and TB [6,29,67] (Figure 3). Thus, the advances made using TEM in unambiguously recognising the distinctive pattern produced during TB degradation of wood make valuable contributions to the research on waterlogged archaeological woods, which can be in a state of decay that does not permit meaningful examination by other forms of microscopy. The wood may be fragile and tunnel remains can only be satisfactorily preserved when such wood tissues are embedded in a suitable polymer prior to sectioning and examination [96].

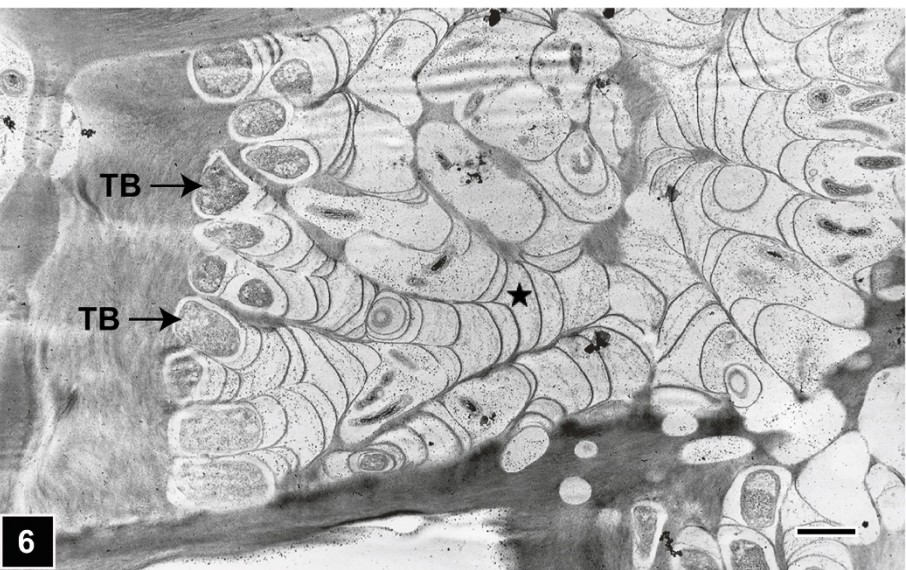

**Figure 6.** TEM of a glancing section through part of a wood cell wall attacked by tunnelling bacteria (TB). Tunnels display repeated branching (asterisk) radiating from a central point. Scale bar = 2 μm. The image is reproduced from Singh et al. (2016) Secondary Xylem Biology, Elsevier.

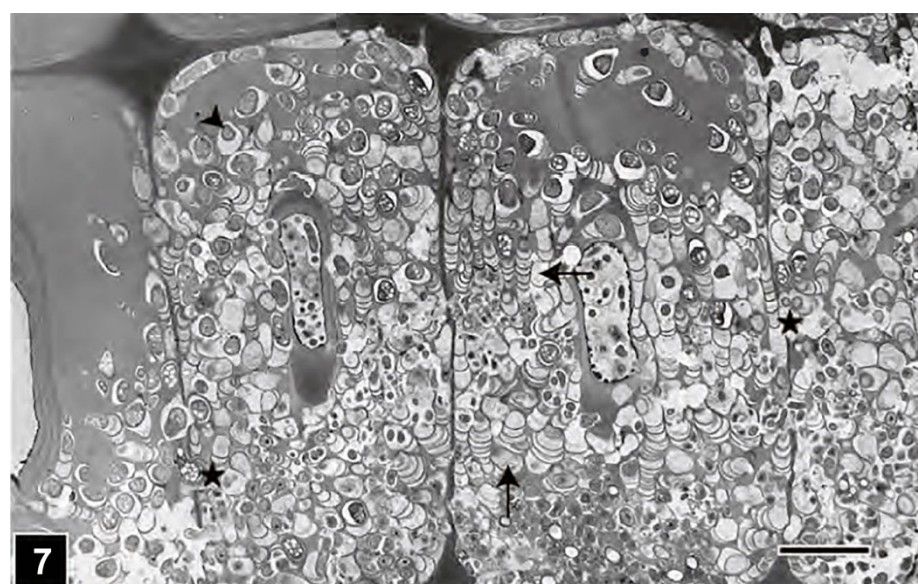

**Figure 7.** TEM of a transverse section through *Homalium foetidum* fibres attacked by tunnelling bacteria (arrowhead). All cell wall regions, including the highly lignified middle lamella (asterisks), are degraded, and the direction of tunnelling (arrows) is variable. Scale bar = 8 μm. The image is reproduced from Singh et al. (1987) J. Inst. Wood Sci.

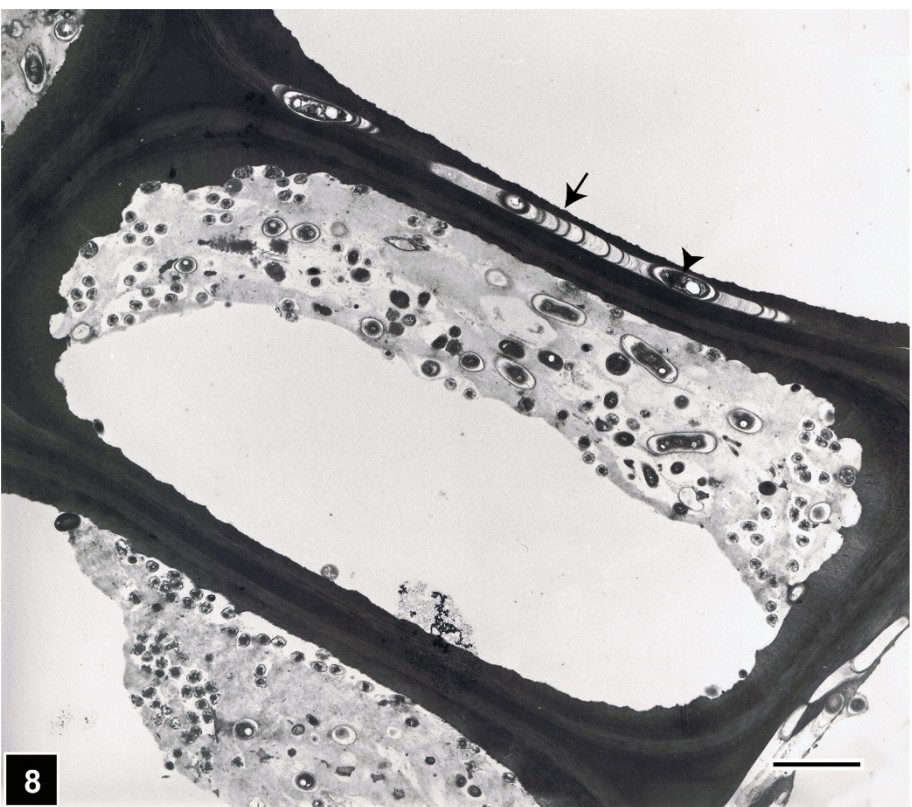

**Figure 8.** TEM of a transverse section through a *Pinus radiata* tracheid attacked by tunnelling bacteria (arrowhead). Tunnelling (arrow) within the S1 layer appears to be along the microfibrils. Scale bar = 8 μm. The image is reproduced from Singh et al. (2019) IAWA J.

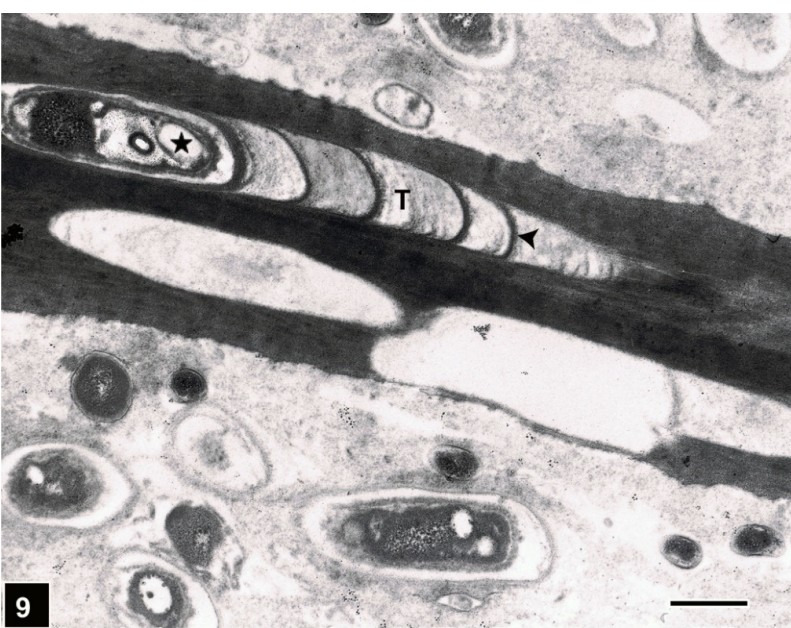

**Figure 9.** TEM of a transverse section through a *Pinus radiata* tracheid wall showing the presence of a tunnel (T) containing a bacterium (asterisk) and crescent-shaped periodic slime bands (arrowhead). Scale bar = 2 μm. The image is reproduced from Singh et al. (2019) IAWA J.

*4.3. Erosion Type Bacterial Degradation*

Advances in understanding bacterial erosion of lignocellulosic cell walls using TEM perhaps have benefitted the research on waterlogged archaeological woods the most, as we now know that such woods are mainly degraded by erosion bacteria (EB) which are considered to be most tolerant to anoxic conditions among wood-degrading microorganisms [16,95,97]. While SEM provided spectacular views of erosion troughs (also called channels), which develop during EB degradation of wood cell walls, and the presence of EB in the troughs [7], it remained for TEM to provide a more precise understanding of the nature of the spatial relationship of EB with cell wall regions under degradation [10,14,98]. TEM also revealed other diagnostic features which are typical of this type of bacterial attack, for example, the presence of cell wall residues (residual material) in degraded cell wall regions [10] and resistance of highly lignified cell wall regions and structures [10,45,47,52–55,98]. Acquisition of high-definition images of the pattern of EB degradation of lignocellulosic cell walls using TEM made it possible to recognise EB type degradation using light microscopy alone based primarily on staining and polarisation characteristics of the residual material [21]. In recent years, this has led to rapid progress in the speedy characterisation of buried and waterlogged archaeological woods using LM, for information on the type of microbial degradation present as well as to obtain samples of wood tissues exclusively attacked by EB for determining chemical changes resulting from EB attack [21]. Further progress made in determining chemical changes due to EB degradation of waterlogged archaeological woods concerns topochemical probing of the residual material and across the cell wall using UV spectrophotometry [99,100] and Raman confocal microscopy [101]. In the context of the above, it is worthy of note that an image (Figure 2 in [82]) from the first LM study of pine wood from a foundation pile undertaken by Walter Liese resembles the degradation pattern described as bacterial erosion.

4.3.1. Environment and Wood Substrates

Like SR and TB, EB are present in a wide range of environments. However, EB commonly occur in wood that is exposed to high levels of moisture and becomes water saturated [52]. In such environments, EB are often present with SR and TB, but the most destructive wood-degrading microorganisms—white-rot fungi and brown rot fungi—are

absent or inactive. For example, EB have been found to co-exist with TB in the same tracheid cells in *Pinus radiata* posts exposed to vineyard soils [91], and with TB and SR in cooling tower timbers [5]. The role of EB in wood degradation becomes more important in environments that lead to complete waterlogging of wood, and consequent depletion of oxygen. It is not therefore surprising that studies undertaken on buried and waterlogged woods [7,8,10], including waterlogged archaeological woods [14,15,19,52,97], have reported the presence mainly of EB attack. It is now well recognised that of all wood-degrading microorganisms, EB are most tolerant to oxygen-depleted conditions [10,14,86,97]. Because wood degradation by EB is rather slow, particularly under anoxic conditions, wooden objects of cultural and historical importance, such as sunken ships and shipwrecks, have been found in a state that can allow preservation/conservation even after hundreds of years of exposure to buried and waterlogged conditions [15,95]. Although in the majority of cases ancient waterlogged archaeological wood tissues have been observed in a heavily degraded state, particularly in the outer layers, wooden Schöningen spears, which had been buried underground, were found to be in excellent condition after 400,000 years because the wood had been attacked only by EB [55]. This provides strong support for the commonly held view that EB degradation of wood under anoxic conditions is extremely slow.

Although EB degradation of both soft and hardwoods has been reported, EB in contrast to TB are not able to degrade cell wall regions that are highly lignified, for example, middle lamella [59,86,95,98] (Figures 10 and 11), initial pit borders in conifers [12,18,45,47], conifer ray tracheids [18,47], highly lignified conifer ray parenchyma [47], warts [47] and the outer S2 wall of compression wood [53,55]. In several studies, EB were found to also degrade CCA-treated timbers placed in service in environments such as cooling towers and in contact with soils, where EB co-existed with SR and TB [4,5,91]. EB degradation of timbers treated with CCA at retention levels high enough to discourage white- and brown rot attack [102] suggests that EB have a high tolerance to toxic chemicals. Degradation of timbers in contact with horticultural soils (oxygenated) [reviewed in 86] as well as buried and waterlogged woods (depleted oxygen) [14,15,18] suggests that EB can be active in a range of environments varying in oxygen concentration.

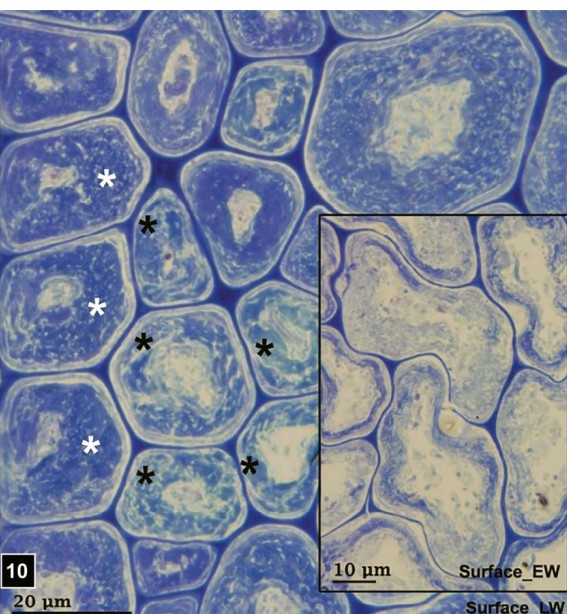

**Figure 10.** LM micrographs of the outer parts (3 cm from the surface) of Daebudo-ship. Section stained with toluidine blue. Latewood (LW) and earlywood (EW) tracheids (inset) contain a granular residue in regions where the secondary cell wall has been degraded. Note the two distinct toluidine blue staining patterns in the LW tracheids (white vs. black asterisks), reflecting compositional variability. The images are reproduced from Cha et al. 2021. IAWA J.

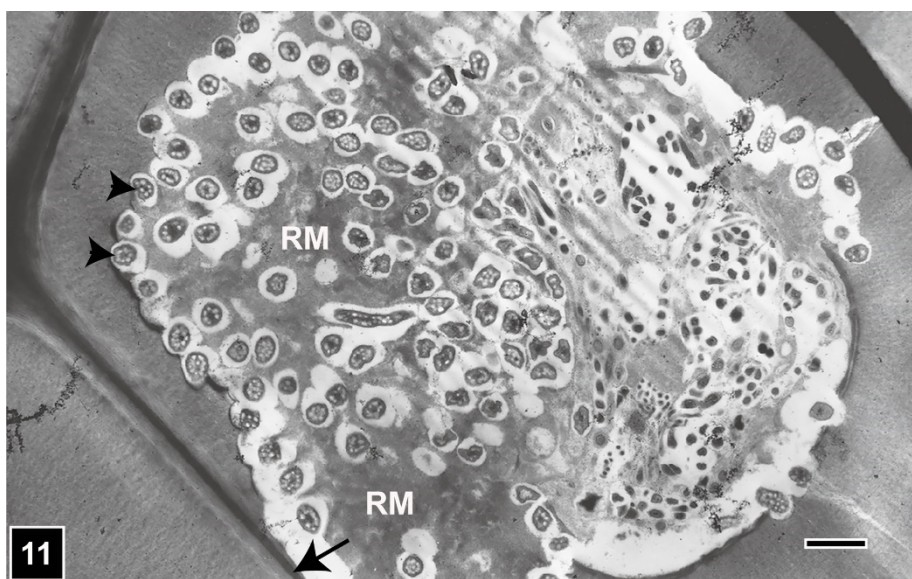

**Figure 11.** Transverse section through Pinus sylvestris tracheid attacked by erosion bacteria (EB). EB are positioned in vicinity to the secondary cell wall, opposite crescent-shaped erosion troughs (arrowheads). The middle lamella is resistant (arrow). The residual material (RM) is dispersed into the lumen, particularly where the S3 has disappeared. Scale bar = 4 μm. The image is reproduced from Singh et al. (2016) Secondary Xylem Biology. Elsevier.

### 4.3.2. Micromorphological Features of EB Degradation: Relevance to Waterlogged Archaeological Woods

Understanding the micromorphology of the pattern produced during bacterial erosion of lignocellulosic cell walls is important for recognising the presence/absence of cell wall erosion caused by bacteria in waterlogged archaeological woods. Advances in unambiguously identifying this type of cell wall degradation have come from the application of SEM and TEM, which provided high-resolution images containing detailed complementary information showing a close spatial association of EB with cell wall regions being eroded, in addition to revealing the form and ultrastructure of these bacteria. The micromorphology of the degradation pattern has been described in several research publications and reviews [10,14,15,29,59,86,95,97]. Like TB, EB are Gram-negative non-flagellate rods, with a membranous cell wall. Although they are similar in size (1.5–2 μm in diameter) to TB, their ends appear conical and thus EB can be distinguished from TB when the two types co-exist, particularly in the lumen of wood tissues where TB display their usual form and are not pleomorphic. EB colonise the cell lumen from where they erode the cell wall generally in the outward direction, i.e., towards the middle lamella. Like TB, EB extrude a slimy material, which facilitates these bacteria to keep in contact with the wood cell wall as the erosion process is initiated and progresses. The degradation process results in a depression into the cell wall facing the bacterium (Figures 11–13), which progressively becomes deeper assuming a distinctive form that has been described as erosion trough [7,10,98]. The channel-like form of erosion troughs is best revealed when viewed with SEM (Figure 12), which is also an ideal tool to examine the form of EB. As viewed with TEM, the channels in transversely cut sections of wood cells appear crescent-shaped cell wall depressions with the EB positioned opposite them, closely fitting into the depressions (Figures 11 and 13) [10,14]. Whereas TB are not generally constrained by the microfibril orientation of cell wall layers and thus are able to move in all directions within the cell wall (Figures 4 and 6), erosion troughs are strictly aligned with microfibrils; in this respect, the behaviour of EB is much like that of SR. In advanced stages of degradation, the coalescence of adjoining troughs results in the loss of their integrity. This feature has relevance to diagnosing the presence of EB attack in waterlogged archaeological wooden artefacts, particularly in the outer heavily degraded regions, where intact troughs may no longer be present.

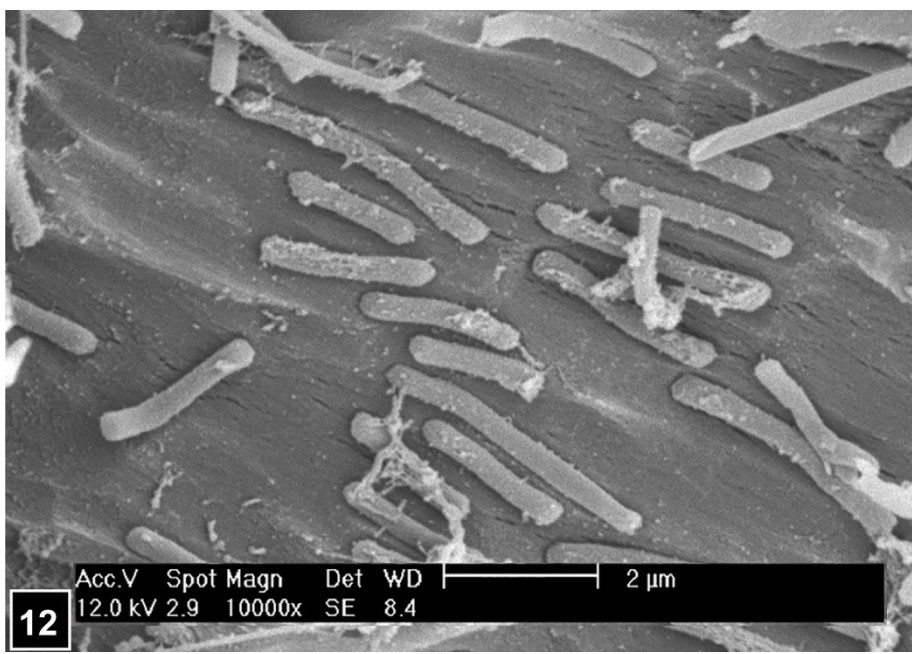

**Figure 12.** SEM of erosion bacteria and underlying erosion troughs produced during cell wall erosion. The micrograph courtesy of Professor Charlotte Björdal, University of Gothenburg, Sweden.

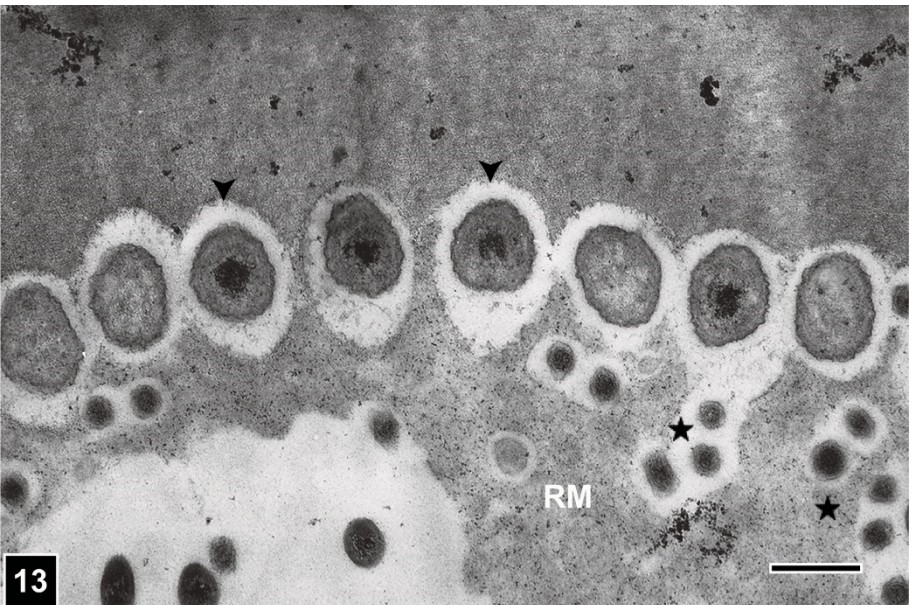

**Figure 13.** TEM of a transverse section through a *Pinus radiata* tracheid wall attacked by erosion bacteria, the primary degraders. The erosion bacteria display a near-circular profile and are present opposite crescent-shaped erosion troughs (arrowheads). The secondary degraders (scavenging bacteria) (asterisks) are associated with the residual material (RM). Scale bar = 2 μm. The image is reproduced from Singh et al. (2019) IAWA J.

However, there are also other characteristic features of EB degradation which are of diagnostic value. Firstly, while in advanced stages, all secondary cell wall regions can be degraded, the lignin-rich middle lamella remains intact (Figures 10, 11 and 14), albeit a loss in the strength of the supporting secondary cell wall can result in distortion of the middle lamella and collapse of wood tissues, particularly when wood is under load or is dried [15,19] (inset in Figure 10).

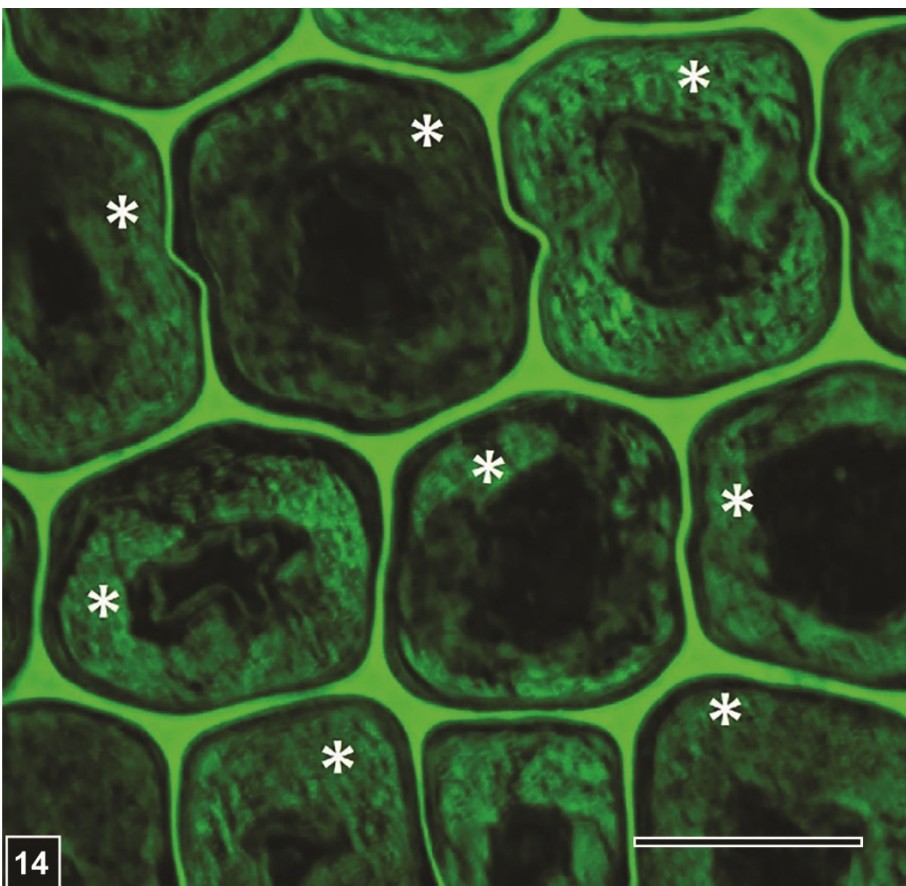

**Figure 14.** Confocal laser scanning micrograph (CLSM) of a transverse section through a waterlogged archaeological wood degraded by erosion bacteria. The residual material (asterisks) displays strong fluorescence for lignin. Scale bar = 15 μm. The micrograph courtesy of Prof. Jong Sik KIM, Chonnam National University, South Korea.

Nevertheless, the presence of middle lamella even in severely degraded wood tissues serves as an important diagnostic feature for EB type degradation. Secondly, as mainly the polysaccharide components of the lignocellulosic cell wall are degraded by EB, the left-over lignin component, which has been described as the residual material (RM), accumulates in the degraded cell wall regions, spreading often into the lumen in the absence of an intact S3 layer (Figures 10, 11, 13 and 14). Following imaging and analysis of the RM by TEM, it has been possible to confirm its presence in EB-degraded cells by LM, viewing sections of degraded wood tissues under polarised light or after staining [21] (Figure 10). Because cellulose is essentially completely lost from degraded cell walls, the lignin-containing RM appears black under polarised light in the absence of birefringent cellulose. These features together with the presence of middle lamella in degraded tissues have proved useful in a rapid assessment of buried and waterlogged archaeological woods from various sites using LM for the presence of EB attack, and in obtaining slices of wood degraded exclusively by EB for chemical analysis [21]. Initially, observations of KMnO4-stained ultrathin sections with TEM indicated the presence largely of lignin in the RM [10] (Figures 11 and 13). In recent years, a range of more specific chemical-based analytical and microscopic methods have confirmed that the RM consists mainly of lignin (Figure 14), which may be slightly altered [21,99–101].

## 5. Understanding Microbial Decay Patterns: Relevance to Preservation of Waterlogged Archaeological Wooden Artefacts

Advances made in understanding the nature and extent of microbial decay of woods from a wide range of waterlogged environments have been a catalyst for renewed inter-

est in appropriately conserving waterlogged archaeological wooden objects based on the knowledge gained [13]. In particular, the following lines of investigations have yielded information of value. First, it is now widely accepted that under conditions that lead to waterlogging of wood, EB attack is the main factor in the deterioration of wood because in waterlogging environments lack of oxygen becomes a limiting factor for wood-degrading microorganisms other than EB, which are extremely tolerant to restrictive oxygen availability [14,86]. It is not therefore surprising that buried and waterlogged archaeological woods have been found to be attacked almost exclusively by EB [14,15]. Second, buried and waterlogged wooden artefacts, such as sunken ships and shipwrecks, have been found in a state that can allow preservation/conservation after hundreds and even thousands of years of exposure, attributable mainly to the very slow rate of wood degradation by EB in anoxic environments. Third, studies of waterlogged archaeological woods have shown that because EB-caused erosion of the cell wall occurs from the surface inwards, wood tissues display various states of deterioration from the extensive degradation of the outmost tissue layers in a wooden object to no degradation of inner tissues, with the presence of tissues in transitional states, some of which even displaying active EB degradation. In any attempt to suitably conserve or restore waterlogged wooden objects that may be highly treasured because of their historical and/or cultural importance, one is advised to take advantage of the above available information. A fitting example where such information has proved valuable is the preservation of the German ship Bremen Cog which was built AD 1380 from Oak wood and was recovered from the river Weser around 1960. An electron microscopic study showed that only the surface layers of the wood components of this ship were degraded by EB and the bulk of the wood was in a sound state [103]. Armed with this knowledge, stabilisation was undertaken, which involved a two-step treatment process, first with PEG (polyethylene glycol) 200 and then with PEG 3000. This process ensured effective impregnation of wood tissues, with PEG 200 impregnating the cell wall of all wood tissues because of its small molecular size, and PEG 3000 impregnating the cell lumen of degraded wood tissues containing highly porous masses of the RM, diffusing into these tissues via their degraded pit membranes. This explains the basis for achieving excellent stabilisation of the ship Cog using PEG of molecular sizes suitable in combination for achieving impregnation of both degraded and sound wood tissues, preventing differential shrinkage that can occur in poorly impregnated waterlogged woods. Fourth, advances made in understanding the texture and chemical composition of the RM present in EB-degraded tissues of waterlogged archaeological woods can serve as a valuable platform for developing suitable stabilisation technologies. TEM studies [10,103] provided an indication that the RM in EB-degraded wood tissues is distinctly more highly porous than sound wood cell walls. In conventional TEM preparations of degraded wood tissues, the dehydrating agents used (acetone, alcohol) can cause shrinkage of cell walls, the shrinkage being much more severe for the highly hydrated residues, such as the RM. Only the techniques, such as cryo-TEM and cryo-SEM [96], can help obtain a more realistic picture of the extent of porosity of the RM and the dimensions of the pores present. Nevertheless, the information available based on the use of conventional TEM techniques is still useful for imaging the texture of the RM, which can form the basis for developing appropriate impregnation formulation, such as that for the stabilisation of the ship Cog [103].

## 6. Future Perspectives

Because wood exposed to waterlogging conditions is mainly attacked by EB, and the residual material that remains subsequent to cell wall degradation occupies a large proportion, particularly heavily degraded tissues, knowledge of the physical and chemical characteristics of this material should bring about improvements in conservation technologies.

More precise assessment of the cell wall porosity across all tissues, from heavily degraded layers to the inner sound wood, will be needed for optimizing the stabilisation of excavated waterlogged archaeological wooden objects. This will require the use of high-resolution tools equipped with facilities to rapidly preserve tissues in as close to their

original state as possible. Cryo-SEM or Cryo TEM are ideal tools for determining the size and distribution of pores in the cell wall from highly degraded and partially degraded (displaying cell walls in varying states of degradation) to sound wood tissues. The process will be rather cumbersome, and only the most valuable artefacts can be targeted, as only two to three layers of tissues will be instantly preserved without the presence of ice crystals which severely distort particularly highly degraded tissues. Here, LM can serve as a useful monitoring tool to obtain suitable thin slices to tissues from various regions for rapid stabilisation prior to microscopic examination and pore size measurement. Atomic force microscopy (AFM), which does not require any prior treatment of tissues, is also a high-resolution tool that has been widely used for imaging cell walls in their native state, particularly to determine the size and arrangement of cellulose microfibrils [104,105]. However, whether this tool would also be suitable for waterlogged wood pieces has to be explored. Other methods [106], including cell wall impregnation with low molecular weight substances, such as PEF [104], have been used. The relevance of such methods for assessing pore size and distribution in EB-degraded archaeological woods should be explored. More precise porosity assessment would be helpful in selecting consolidants of appropriate molecular sizes and determining their combination and sequences for the treatment of recovered artefacts. Knowing that RM primarily consists of modified lignin, it is tempting to suggest that a consolidation technology can be developed based also on the precise chemical composition of RM, with a view to finding agents that can be grafted on the components of the RM in a watery medium.

Although several studies have analysed the microbial community present in waterlogged archaeological woods [107–110], we still do not know the true identity of EB, the main degraders of buried and waterlogged woods [14,15,17]. Knowing that it is possible to obtain slices of wood from regions exclusively degraded by EB, as monitored using LM [21], specific tissue regions from waterlogged wooden objects under active degradation can serve as a suitable material for determining the taxonomic affiliation of EB using well-tested molecular biological tools and techniques [111]. Although TEM has revealed the frequent presence of scavenging bacteria in wood cells being degraded by primary degraders EB in anoxic environments [10,86], an analysis should still be possible as EB biomass will serve as the main source of material for RNA/DNA-based molecular analysis when tissue slices can be taken from the region of active degradation.

**Author Contributions:** A.P.S. conceptualised the review paper. Y.S.K. and R.R.C. contributed to designing the frame of review paper. A.P.S. wrote the manuscript. Y.S.K. and R.R.C. contributed to enhance the readability of the manuscript and the quality of figures. A.P.S. and Y.S.K. revised the manuscript. All authors have read and agreed to the published version of the manuscript.

**Funding:** This research received no external funding.

**Acknowledgments:** Our grateful thanks to Charlotte Bjordal and Jong Sik Kim for kindly providing micrographs for the figures (Figure 12 and Figures 1 and 14, respectively), and the anonymous reviewers for their helpful comments and suggestion.

**Conflicts of Interest:** The authors declare no conflict of interest.

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
