# Peer review of "Advances in Understanding Microbial Deterioration of Buried and Waterlogged Archaeological Woods: A Review"

_forests, doi:10.3390/f13030394_

Round 1

Reviewer 1 Report

The article is well prepared and gives complex information about archeological woods and the techniques to analyzed thad type of material.

Author Response

Reviewer 1

Suggestion

Minor spell check required

Response

Spell check done. Please note that we have used British English (and not American) spelling throughout.

Reviewer 2 Report

This paper is a very interesting review on the bacterial degradation of wood and in particular on the bacterial degradation of waterlogged archaeological wood.

The review of the scientific literature is quite accurate, even if there are no references to the work done on the grading of micromorphological decay, that is, the decay evaluated on the basis of the attack level of the cell walls.

It should also be emphasized in the introduction that the anatomical assessment of decay is one, but not the only, step to obtain a complete diagnosis.

The type of decay of waterlogged archaeological wood due to bacteria is also the main cause of the "paradox of the waterlogged archaeological wood" artefacts. They often deceives archaeologists in the digging phase. Since there is an erosion of the cell wall from the lumen, that often maintains the shape of the individual wood cells (thanks to the middle lamella), the presence of the turgor given by the water also maintains the shape of the artifact, which as soon as it emerged from the excavation appears in an excellent state of conservation (lines 493-496), while in reality the material can be heavily degraded, up to a level that makes it difficult the anatomical wood identification.

In my opinion there are some shortcomings in paragraphs 5 and 6.

Firstly: only PEG is mentioned as material for consolidation. The diagnosis, together with other information, may also suggest other materials and other methodologies for consolidation, in some cases just a controlled drying.

Secondly: but is it only the porosity of the material that suggests how to consolidate? I believe the residual chemical composition is the same, together with physical evaluations (residual density?).

Thirdly, the role of lignin: it is defined as "hygroscopic" and "modified". These are aspects that concern a few more lines. Why hygroscopic a material that is normally defined as “hydrophobic”; thus: how modified?

Finally: are we sure that cryo-SEM can be a useful tool? Ice normally causes a shattering of the degraded walls, in addition to the collapse of the tissues. Are there any experiences in literature?

Author Response

Reviewer 2

Statement

Lines 493-496 -  '--while in reality the material can be heavily degraded--'

Response

Agree. In archaeological wood exposed to waterlogging conditions for prolonged periods, attacked by erosion bacteria, are indeed heavily degraded, particularly in the outermost layers of the wooden artefacts, giving a false impression that the excavated wood is well preserved, based on the near-normal morphological appearance of degraded wood tissues.

Under 4.3.1. Environment and Wood Substrate

Line 491 - Original statement 'have been found to be relatively well preserved' re-stated as 'have been found in a state that can allow preservation/conservation'

Line 493 - The sentence modified as 'Although in majority cases ancient waterlogged archaeological wood tissues have been observed in a heavily degraded state, particularly in the outer layers, wooden Schöningen--, which had been attacked by erosion bacteria, --'

Comment

Firstly, only PEG mentioned as material for consolidation.

Response

The main aim of the review was to present information on the advances made in understanding biodegradation of archaeological wood. Therefore, conservation was only briefly mentioned, giving an example of the use of PEG as a widely used consolidant, with the expectation that the special issue of the journal Forests entitled 'Waterlogged Arcaheological Woods' will attract manuscripts with an in-depth coverage of conservation methods.

In regard to use of other conservation methods, reference was made to a review paper detailing use of conservation methods. This reference (Jiang et al. 2018) has now been deleted and replaced with a more recent review paper (Broda and Hill 2021), which gives a more comprehensive and excellent coverage of conservation process and the methods used for conservation.

Comment

Secondly, is it only the porosity of the material that suggests how to consolidate?

Response

Agree, it is not only the porosity that should be considered as an important parameter in undertaking conservation of waterlogged archaeological wooden objects, and there are several other aspects of such wooden artefacts that have to be considered, for example the characteristics of inner undegraded tissues, cell wall features of tissues in transitional states, and the characteristics of extensively degraded outer tissues. The intent was to bring into consideration residual material, which forms the bulk material of the heavily degraded tissues and which per se has not been specifically mentioned in relation to conservation. The following lines have been now inserted in the manuscript.

Future Perspectives

The opening lines under this heading now read 'Because wood exposed to waterlogging conditions is mainly attacked by EB, and the residual material that remains subsequent to cell wall degradation occupies a large proportion particularly of the heavily degraded tissues, a knowledge of the physical and chemical characteristics of this material should bring about improvements in conservation technologies.'

Comment

Thirdly, the role of lignin: it is defined as "hygroscopic" and "modified".

Response

The residual material was not described as being hygroscopic, instead it was stated that because it is present in a watery environment, selection of consolidant(s) should be based on the chemical composition and its porosity and the environment it is present in.

Waterlogged wood present in anoxic or oxygen-depleted environments are often heavily degraded by erosion bacteria, which are the main factor for the deterioration of this type of wood, contains large amounts of residues (residual material) left over in degraded cell wall regions, spreading also in the cell lumen. Many physical and anatomical characteristics have been taken into consideration during conservation of waterlogged wooden objects, and in this review another important factor, the residual material which forms the bulk component of heavily degraded wood tissues and which is thought of consisting largely of lignin and/or modified lignin, is proposed to be considered. Still, the exact chemical composition of this material and its chemical and physical heterogeneity, including its density and porosity, are not well understood.

Comment/question

Finally, are we sure that cryo-SEM can be a useful tool?

Response

To our knowledge, cryo-SEM and cryo-TEM have not been used to examine the state of degraded wood tissues in waterlogged archaeological wood, and therefore we are recommending these tools and techniques particularly to assess the porosity of the cell wall residue (residual material) that forms the bulk of the material component of waterlogged archaeological wood tissues heavily degraded by erosion bacteria. Therefore, effective stabilization of this material using appropriate methods is important. However, prior to use of a consolidant (or a combination of consolidants) it is important to assess the porosity (size and distribution of pores) of the residual material. Cryo-SEM and/or cryo-TEM in combination with rapid freezing (e.g., liquid propane cooled by liquid nitrogen), can minimize distortions caused by ice crystals that form during conventional preparations involving freezing. Furthermore, modern electron microscopes permit imaging at high resolution at a low kV. So, our intent was to encourage exploration of these technical approaches novel to the study of waterlogged archaeological wood. However, the techniques proposed are rather tedious, and because only extremely small size samples can be rapidly frozen at very low temperatures, the recommended technique should be tried on only the most precious artefacts.

Reviewer 3 Report

„Advances in Understanding Microbial Deterioration of Buried and Waterlogged Archaeological Woods: A Review” is an interesting article thoroughly presenting all issues related to microbial degradation of waterlogged wood. It will be interesting for both wood scientists and wood conservators. I suggest adding some more information to particular paragraphs – more detailed comments can be found in the attached pdf file.

Author Response

Suggestion

Minor spell check required

Response

Spell check done. Please note that we have used British English (and not American) throughout.

Suggestion

Lines 40-45. I would suggest to divide this paragraph into shorter parts - separately about degradation of waterlogged wood, separated about the need to preserve such wooden artefacts (line 52 - a new paragraph).

Response

Divided into two paragraphs as suggested (lines 40-52) and (lines 52-63).

Comment

Line 89. Where: in soft wood? Please specify.

Response

Line inserted as 'in both soft- and hardwoods [e.g., 28]

Comment/suggestion

Line 207. Mentioning Type 1. I suggest there are also other types of soft rot degradation, but the authors do not mention them in the manuscript. I suggest adding an explanation about the types of soft rot and why only Type 1 has been described here.

Response

Following lines inserted under 3. Fungal Degradation:

Soft rot fungi cause two well defined patterns of wood degradation, which have been defined as Type I and Type II. Type I is characterized by cavity formation within the cell wall during the degradation process. In Type II, the cell wall is eroded by fungal hyphae present in the cell lumen. In our review, the micromorphological pattern of only soft rot Type I is presented, as it has been reported that Type I soft rot is exclusively present in waterlogged archaeological wood.

Comment/question

Line 451. What about cavitation bacteria? There is some information about this type of microbial attack of waterlogged wood in the literature - please provide some information about it.

Response

Although still being reported in some review papers [e.g., Broda and Hill 2021], it has not been possible to reproduce cavitation type bacterial decay originally described by Nilsson and Singh [1984 -IRG Document No. 1235] in laboratory studies. The initial description was based on the presence of what appeared to be cavities in the cell wall, but subsequent studies have shown that degradation of the cell wall by erosion bacteria deep within the S2 layer gives an appearance of the presence of cavities when viewed under a light microscope, and particularly in regions where S3 is present overlying cavity-type formations. However, transmission electron microscopic observations have provided evidence of the presence of a residual wall material and bacteria opposite well defined depressions (erosion troughs) into the cell wall. Based on the above, it is now considered by the leading experts in the field of wood biodegradation that cavitation is indeed a form of bacterial erosion, and should not be placed in a separate category.

Therefore, cavitation type bacterial attack is not described in our review.
